# Chromone-Containing Allylmorpholines Influence Ion Channels in Lipid Membranes via Dipole Potential and Packing Stress

**DOI:** 10.3390/ijms231911554

**Published:** 2022-09-30

**Authors:** Svetlana S. Efimova, Vera A. Martynyuk, Anastasiia A. Zakharova, Natalia M. Yudintceva, Nikita M. Chernov, Igor P. Yakovlev, Olga S. Ostroumova

**Affiliations:** 1Institute of Cytology of Russian Academy of Sciences, Tikhoretsky 4, 194064 Saint Petersburg, Russia; 2Saint-Petersburg State Chemical Pharmaceutical University, Professor Popov 14, 197022 Saint Petersburg, Russia

**Keywords:** allylmorpholine, chromone, membrane, ion channel

## Abstract

Herein, we report that chromone-containing allylmorpholines can affect ion channels formed by pore-forming antibiotics in model lipid membranes, which correlates with their ability to influence membrane boundary potential and lipid-packing stress. At 100 µg/mL, allylmorpholines **1**, **6**, **7**, and **8** decrease the boundary potential of the bilayers composed of palmitoyloleoylphosphocholine (POPC) by about 100 mV. At the same time, the compounds do not affect the zeta-potential of POPC liposomes, but reduce the membrane dipole potential by 80–120 mV. The allylmorpholine-induced drop in the dipole potential produce 10–30% enhancement in the conductance of gramicidin A channels. Chromone-containing allylmorpholines also affect the thermotropic behavior of dipalmytoylphosphocholine (DPPC), abolishing the pretransition, lowering melting cooperativity, and turning the main phase transition peak into a multicomponent profile. Compounds **4**, **6**, **7**, and **8** are able to decrease DPPC’s melting temperature by about 0.5–1.9 °C. Moreover, derivative **7** is shown to increase the temperature of transition of palmitoyloleoylphosphoethanolamine from lamellar to inverted hexagonal phase. The effects on lipid-phase transitions are attributed to the changes in the spontaneous curvature stress. Alterations in lipid packing induced by allylmorpholines are believed to potentiate the pore-forming ability of amphotericin B and gramicidin A by several times.

## 1. Introduction

The regulation of cholinergic and glutamate neurotransmission represents a main approach in the symptomatic treatment of the cognitive and behavioral symptoms of Alzheimer’s disease. In this regard, the available pharmacological treatments involving the use of acetylcholinesterase (AChE) inhibitors and *N*-methyl-*d*-aspartate (NMDA) receptor antagonists are still of current interest. The multifactorial nature of Alzheimer’s disease and its various cerebral mechanisms of pathogenesis determine the necessity of creating multitargeted drugs [1]. In this regard, morpholine derivatives have large perspectives for clinical application. The morpholine ring is frequently applied for the synthesis of compounds with multiple psycho-neurological activities, including inhibitors of AChE and NMDA receptors [2,3,4,5,6]. A recent study reported that chromone-containing allylmorpholines show high selectivity toward AChE, which is combined with the moderate antagonism of NMDA receptors [7]. This makes morpholine derivatives a perspective group for drug development for the treatment of neurodegenerative diseases. Although the lipophilicity of morpholine-containing molecules largely depends on their functional groups, the ring by itself offers a well-balanced lipophilic–hydrophilic profile with desirable drug-like characteristics [8]. LogP of morphoine derivatives usually reaches 3–4, demonstrating satisfying lipophilicity that might be sufficient for the modulation of the physicochemical properties of the lipid bilayer of cell membranes and reconstituted proteins and peptides [9].

AChE and NMDA receptors are membrane-associated proteins, and modulation of the lipid matrix can affect their conformational transitions and functioning. As such, it was shown that phase transition in membrane lipids affects AChE activity [10]. A general anesthetic, ketamine, which has a disordering effect on membranes, was also shown to regulate AChE functioning. Mazzanti et al. [11] reported that the AChE of synaptosomes from the brains of anaesthetized rats displayed decreased specific enzymatic activity in response to increased fluidity in the membrane when compared to synaptic membranes from the animals not treated with ketamine. Curatola et al. [12] also investigated the effects of ketamine on the activity of the human erythrocyte enzyme. The inhibitory action of the anesthetic was lost after the solubilization of AChE by sonication, and it was restored by enzyme reconstitution into a lipid environment, demonstrating that the effect was mediated by membrane lipids. In turn, NMDA receptors respond to the modulation of elastic membrane properties via lysophospholipids and arachidonic acid. The introduction of lysophospholipids with a large hydrophilic headgroup resulted in the inhibition of NMDA-mediated responses, whereas arachidonic acid with a small hydrophilic headgroup potentiated NMDA receptor currents [13]. These facts might indicate that perturbations in the lipid microenvironment, especially during curvature stress, are able to alter the receptor location and function of AChE and NMDA.

The aim of this study was to rationalize whether the effects of chromone-containing allylmorpholines on the lipid microenvironment could partially underlie the regulation of the ion channels incorporated into model lipid membranes. To achieve this goal, we revealed the alterations induced by chromone-containing allylmorpholines in the distribution of the electrical potential at the membrane–aqueous interface and the thermotropic behavior of membrane lipids. We also showed that agent-induced amendments in the bilayer properties modulated the functions of ion channels produced by gramicidin A and amphotericin B. The lipid-mediated action of the tested compounds related to the modulation of both the electrical and elastic properties of lipid bilayers should be taken into account when studying the effects of allylmorpholine derivatives on various membrane proteins.

## 2. Results and Discussion

### 2.1. The Effects of Chromone-Containing Allylmorpholines on the Transmembrane Distribution of Electrical Potential

#### 2.1.1. Membrane Boundary Potential

The drop in electrical potential at the membrane/aqueous interface, the membrane boundary potential (*φ_b_*), consists of two components: surface (*φ_s_*) and dipole (*φ_d_*) potential [14,15,16,17]. The first component refers to a diffuse part of the electrical double layer and to the screening of the membrane’s surface charge by ionic media. The second is attributed to the specific orientation of the lipid and water dipoles at the interface and depends on lipid hydration and packing as well as on the intercalation of small molecules [18].

Appendix A shows the dependences of the changes in the boundary potential of the POPC bilayers (Δ*φ_b_*) on the concentration of chromone-containing allylmorpholines in a bathing solution (*C*). The variations in the chemical structures of the tested chromone-containing allylmorpholines are presented in Table 1. In the concentration range of 10–50 μg/mL, the potential-modifying effect of the tested compounds increased linearly as the dose increased. At the concentrations of 50–200 μg/mL, the presented curves tended to show saturation. This fact implied that the surface density of the adsorbed molecules was at about the saturation limit at these concentrations. The concentration at which the saturation limit was reached and the maximal values of the changes in *φ_b_* were strictly dependent on the allylmorpholine type. Table 1 presents the values of the changes in *φ_b_* in the presence of various chromone-containing allylmorpholines at 100 μg/mL. The efficiency of reducing the membrane boundary potential decreased in the series **1** ≈ **6** ≈ **7** ≈ **8** (Δ*φ**_b_* was about −100 mV) > **3** ≈ **10** ≈ **11** ≈ **12** (Δ*φ**_b_* varied in the range of −70 to −50 mV) > **2** ≈ **9** (Δ*φ**_b_* was equal to 25 mV). The derivatives **4** and **5** led to insignificant increases in the membrane boundary potential at 20–25 mV.

Next, we decided to find out which component, the surface or the dipole potentials, was responsible for the changes in the boundary potential during the adsorption of alylmorpholines on the membrane.

#### 2.1.2. Membrane Surface Potential

Less than 10% of chromone-containing allylmorpholine molecules are in the ionized form at pH 7.4 (the ionization constants are presented in Appendix A); therefore, it is unlikely that the addition of allylmorpholines caused a change in the surface charge of the membranes. In order to exclude the possibility of the ionization of allylmorpholines in the lipid microenviroment, we measured the zeta-potential of the POPC liposomes in the absence and presence of the tested compounds using the dynamic light scattering technique [16]. The zeta-potential is believed to be an important and reliable indicator of the surface charge of membranes, and, consequently, of their surface potential. Table 1 demonstrates the values of the *ξ*-potential of the POPC vesicles treated with the tested allylmorpholines (*ξ*-potential of liposomes modified by different compounds varied in the range from −4 to 0 mV). The zeta-potential of the untreated POPC vesicles was equal to −2.1 ± 0.7 mV. Thus, the data obtained did not indicate any noticeable changes in the surface potential of membranes upon allylmorpholine adsorption, and the observed allylmorpholine-induced changes in the membrane boundary potential could be attributed to its dipole component being altered by the uncharged form of the allylmorpholine molecules.

#### 2.1.3. Membrane Dipole Potential

To clearly demonstrate the key role of dipole potential modulation in the action of allylmorpholines, we performed fluorescence intensity measurements using a dipole potential-sensitive lipid fluorescence probe, di-8-ANEPPS. The results obtained indicated a high dipole-modifying ability of **1**, **3**, **6**, **7**, **8**, **10**, **11**, and **12** (Δ*φ_d_* was equal to 60 ÷ 120 mV) (Table 1). Moreover, the Δ*φ_b_* and Δ*φ_d_* values were close, indicating the prevailing role of the dipole component in the changes in the membrane boundary potential under allylmorpholine action.

The logarithms of the octanol/water partition coefficients (*LogP*) are presented in Appendix A. No correlation between the *LogP* and │Δ*φ_b_*│ values of the POPC bilayers was observed. This fact indicates that the dipole-modifying action of the agents is not directly related to the lipophilicity of the molecules. At the same time, there was a correlation between the magnitudes of the dipole moments of the allylmorpholine molecules (the *μ* values are presented in Appendix A) and the │Δ*φ_b_*│ values (Pearson correlation coefficient is equal to 0.67), indicating the greater importance of molecule polarity.

The relationships between the structures of the allylmorpholines and their potential-modifying abilities (Table 1) were analyzed, and several important observations were made:

(i) Increasing in the length of the side hydrocarbon chain/chains (R_5_ or both R_4_ and R_5_) in the series of **8** < **11** < **9** (Figure 1A) and **7** < **12** (Figure 1B) led to a pronounced drop in the potential-modifying efficiency. Taking into account the fact that the magnitude of the changes in the dipole potential depends on the normal component of the projection of the dipole moment of the modifying molecule, one possible explanation implies the reorientation of the polar part of the allylmorpholine molecules at the immersion of highly hydrophobic R_4_ and R_5_ into the membrane’s hydrocarbon core.

(ii) The type of halogen in the R_1_ position (Br, F, and Cl in **1**, **3**, and **7**, respectively) did not greatly affect the Δ*φ_b_* value (Figure 1C). Moreover, the presence of an electron-withdrawing substituent in this position is a determining factor: the non-substituted derivative (**2**) hardly exhibits dipole-modifying activity compared to the halogenated analogues (**1**, **3**, **7**) or to the compound with a NO_2_ group (**6**).

(iii) The replacement of the electron-withdrawing substituent in the R_1_ position (**1**) for a hydrophobic methyl radical and the incorporation of electronegative bromine into the R_3_ position (in the case of compounds **4** and **5**) probably changes the orientation of the allylmorpholine molecule in the bilayer in such a way that compounds **4** and **5** increase the dipole potential instead of its reduction, as caused by **1** (Figure 1D).

#### 2.1.4. Gramicidin A channels

The membrane’s dipole potential affects the pore-forming ability of various antimicrobial agents, including gramicidin A (GrA) [19,20,21,22,23,24,25], alamethicin [26,27,28], syringomycin E [29,30,31], surfactin [32], cecropin A [33], and HPA3 peptide [34]. Dipole potential also influences the activity of OmpF porin [35]. An alteration in the dipole potential by small molecules has also been shown to potentiate a voltage-gated human ether-a-go-go-related gene potassium channel (hERG) [36].

GrA is a well-known pore-forming peptide antibiotic produced by *Bacillus brevis.* It is believed that two GrA monomers in opposite monolayers form a transmembrane dimer with conductive properties. The conductance of single GrA channels is highly sensitive to changes in bilayer dipole potential [19,20,21,23,24,25,31]. The almost ideal cation permeability of the GrA channel [37,38] makes it an excellent model to investigate the electrostatic effects: the decrease in dipole potential (with the hydrocarbon region being positive relative to the aqueous phase) is expected to diminish the electrostatic energy at the center of the pore for cations [39], i.e., to increase channel conductance.

Figure 2A shows the current fluctuations induced by opening and closing single GrA channels in the POPC membranes at a transmembrane voltage of 150 mV in the absence (*control*) and presence of derivatives **5** and **7** at the concentration of 100 μg/mL. Compound **5** did not affect the channel amplitude, whereas derivative **7** markedly increased GrA channel conductance compared to the control value (Figure 2A). The effects of allylmorpholines on GrA pore amplitude at 150 mV are summarized in Table 1. An addition of **1**, **3**, **6**, **7**, **8**, and **12** significantly enhanced GrA channel conductance by about 10–30%. The shape of the *g_sc_* (*V*) curves of the GrA channels was not influenced by an addition of allylmorpholines (Figure 2B). The high negative value (−0.84) of the Pearson correlation coefficient between the *g_sc_^agent^/g_sc_^control^* and Δ*φ_b_* values (Table 1) agrees with the strong negative correlation and the exponential growth of the cation conductance when the membrane dipole potential decreases [39]. The relatively small observed changes in the channel conductance (10–30%) are consistent with the substantial shielding of the membrane dipole potential in the interior of a gramicidin pore [40]. Thus, we assumed that chromone-containing allylmorpholines affect the conductance of GrA channels via modulation of the membrane dipole potential (Figure 3). The addition of compound **7** diminished membrane dipole potential and caused a decrease in the cost of cation translocation through the GrA channel.

Figure 2C demonstrates one more effect of the chromone-containing allylmorpholines on GrA channels: the introduction of compound **7** was accompanied by a sharp increase in peptide pore-forming activity. Agent **5** was not able to produce a similar effect. Table 1 presents the product of the number of GrA channels and their probability of being open (*NP_op_*) to characterize the changes in the channel-forming activity of GrA in the presence of the tested allylmorpholines. *NP_op_* was enhanced by 3–30 times upon the addition of **1**, **3**, **4**, **6**, **7**, **8**, **10**, **11**, and **12**. This effect might be attributed to prolonging the channel lifetime (and, consequently, *P_op_*) at the compound-induced reduction in *φ_d_* [19,20,22]. The dipole potential is believed to affect GrA channel lifetime via the influence on the movement of the polar groups of GrA molecules through the region experiencing dipole potential drop at the membrane–aqueous interface [19]. Thus, the observed growth in *NP_op_* at the adsorption of derivatives decreasing *φ_d_* (**1**, **3**, **6**, **7**, **8**, **10**, **11**, and **12**) (Table 1) was in agreement with an assumption that chromone-containing allylmorpholines affect GrA’s pore dwell time by decreasing dipole potential drop. However, the smaller but noticeable three-fold increase in the *NP_op_* value in the presence of **4** did not satisfy this concept, demonstrating a more complex nature of the modulation of GrA’s pore-forming ability by allylmorpholines. The formation of GrA channels by the dimerization of peptide subunits in opposite lipid leaflets causes local compression and bending due to a mismatch between the GrA channel’s hydrophobic length and the thickness of the hydrocarbon core of the unmodified membrane [22,41]. Therefore, *NP_op_* is affected by the bilayer thickness and lipid-packing stress [22,41,42,43,44]. Thus, an increase in the pore-forming ability of GrA in the presence of allylmorpholines might imply alterations not only in the electrical but also in the elastic properties of the bilayers.

### 2.2. The Effect of Chromone-Containing Allylmorpholines on Lipid Packing

#### 2.2.1. Thermotropic Behavior of Membrane Lipids

To address the question about the action of chromone-containing allylmorpholines on lipid packing, we employed differential scanning microcalorimetry measurements. Figure 4 shows the representative thermograms of DPPC before and after the addition of up to 100 µg/mL of allylmorpholines into a liposome suspension. The tested compounds were shown to affect the thermotropic behavior of DPPC. Compounds **1**, **2**, **3**, **4**, **6**, **7**, **8**, **10**, **11**, and **12** completely abolished the gel-to-ripple phase transition, while **5** (Figure 4A) and **9** (Figure 4B) shifted the pretransition peak towards higher temperatures by 0.8 and 0.5 °C, respectively (Table 2). These results suggest that the membrane binding of allylmorpholine derivatives altered the delicate geometrical balance between the lipid heads and chains [45]. Allylmorpholines also decreased the sharpness of the main (gel-to-liquid) phase transition of DPPC (Figure 4A,B). The changes in the temperature difference between the onset and completion boundaries of melting (∆∆*T_b_*) are given in Table 2 (the peak was expanded by 1.1–3.9 °C compared to pure DPPC). High values of ∆∆*T_b_* indicate a decrease in the cooperativity of the main transition of DPPC in the presence of the tested compounds. This was also expressed by the 1.1–3.2-times drop in the cooperative unit size (Table 2 presents the ratio of the *CUS* values in the absence and presence of chromone-containing allylmorpholines).

The introduction of **4**, **5**, **9**, **10**, **11**, and **12** turned the main peak into a multicomponent profile consisting of two or three overlapping transitions, while **1**, **2**, **3**, **6**, **7**, and **8** did not produce a noticeable multi-phase transition at 100 µg/mL (Appendix A). The number of components fitting DPPC’s melting profile increased, and deconvolution was observed for all of the tested agents at 250 µg/mL (Appendix A). Good profile matching resulting from repeated heating steps (Appendix A) indicated that the complex nature of the main peak was not a result of a non-equilibrium distribution of the derivatives between lipid monolayers. Appendix A presents the results of the decomposition/deconvolution analysis of the main DPPC transition peak in the presence of allylmorpholines. It shows that if the number of components fitting the main peaks at 100 and 250 μg/mL coincided, the percentage contribution in the total area of the peak with highest (*T_m_1_*) and lowest melting points (*T_m_2_*/*T_m_3_* in the case of two/three overlapping transitions) decreased and increased, respectively. Thus, the components with different melting points should be attributed to the melting of practically pure DPPC and allylmorpholine-enriched regions, which might be consistent with the interdigitated and non-interdigitated lipid domains [46]. The latter assumption was supported by the typical loss of the pretransition (Table 2) and an increase in *T_m_* hysteresis (the difference in the melting temperature between heating and cooling scans), with an increase in the concentration of agents (Appendix A). Table 2 and Appendix A demonstrate that **4**, **6**, **7**, and **8** were also able to decrease *T_m_*__1_ at the even lower tested concentration of 100 μg/mL (compared to its control value of 41.5 °C), while in the presence of other compounds, *T_m_*__1_ was equal to the magnitude of the untreated membrane. This might indicate the disordering action of **4**, **6**, **7**, and **8** on DPPC bilayers.

According to [47], the modification of the gel-to-liquid-crystalline phase-transition profile by the tested compounds (which typically reduced the cooperativity and the formation of multi-phase transition, growing the relative contribution of the component with lowest melting point by increasing the agent concentration and resulting in an appearance of additional extrema) identified chromone-containing allylmorpholines as type B/D additives, which are usually located between the membrane surface and the hydrophobic–hydrophilic interface of the bilayer. The assumed localization of the tested compounds might be accompanied by the induction of positive spontaneous curvature stress.

In order to show that allylmorpholines can affect curvature stress, we performed differential scanning microcalorimetry with POPE. Cone-shaped POPE is known to form an inverted hexagonal phase, and the low-enthalpy transition from a lamellar to inverted hexagonal phase (H_II_) can be detected using highly sensitive microcalorimetry [48,49]. We showed that derivative **7** (exhibiting a *T_m_*__1_-decreasing effect on DPPC) was also capable of shifting the peak corresponding to the transition of POPE from a lamellar to H_II_ phase when temperatures increased by more than 5 °C when introduced into liposome suspension at the lipid:compound ratio of 10:1 (Appendix A). According to [50,51,52], this fact indicates the production of positive spontaneous curvature stress by the compound, whereas the opposite effect (a decrease in *T_H_**_II_*) should be interpreted as the induction of negative curvature stress by membrane-modifying molecules [48,49,53].

#### 2.2.2. Amphotericin B Channels

Many pore-forming agents and ion channels are sensitive to changes in the spontaneous curvature stress [41,54,55,56,57,58]. In particular, antifungal polyene macrolide antibiotics, such as amphotericin B (AmB) and nystatin, are introduced from one side of the sterol-enriched membrane, forming asymmetric pores. The most widely accepted polyene channel model implies the formation of polyene–sterol complexes that are associated with a barrel-type structure [59], but their lengths are not long enough to penetrate the entire bilayer [60,61,62]. The induction of local positive curvature stress decreases the cost of pore formation by polyenes and increases their pore-forming activity. Thus, the single-length (one-sided) pores produced by polyenes are known to be sensitive to chemically induced changes in membrane elastics [23,63].

Figure 5 presents the effects of **4**, **6**, **7**, and **8** on the steady-state transmembrane current induced by the one-side addition of AmB. Appendix A shows the kinetics of an AmB-induced transmembrane current in the presence of other tested compounds. Table 2 summarizes the mean ratios of the steady-state transmembrane currents induced by AmB in the presence and absence of allylmorpholines at 100 µg/mL (*I_∞_^agent^/I_∞_^control^*). The addition of **4**, **6**, **7**, and **8** led to an increase of 1.4–3.4 times in the steady-state AmB-induced transmembrane current, while the other tested derivatives practically had no effect on AmB’s pore-forming activity (*I_∞_^agent^/I_∞_^control^* was about 1).

Data from the calorimetric study (such as the type of changes in DPPC’s melting profile in the presence of allylmorpholines, the decreased *T_m_*__1_ value of DPPC caused by **4**, **6**, **7**, and **8**, and the inhibition of the formation of an inverted hexagonal phase in POPE by **7**) as well as the results of the electrophysiological measurements demonstrating the growth in the pore-forming ability of GrA in the presence of **4** and of AmB in the presence of **4**, **6**, **7**, and **8** do not contradict the assumption that allylmorpholines induce positive curvature stress. Thus, we assumed that chromone-containing allylmorpholines affected the pore-forming activity of AmB via alterations in the lipid-packing stress. (Figure 6). The two-sided addition of the derivative **7** to the membrane-bathing solution decreases the high cost of formation of the lipid mouth of the AmB pore as the positive curvature and the number of AmB channels increases.

## 3. Materials and Methods

### 3.1. Chemical Reagents

Nonactin, KCl, HEPES, phosphate-buffered solution (PBS), EDTA, pentane, ethanol, calcein, triton X-100, sephadex G-50, DMSO, di-8-ANEPPS, PEG-8000, amphotericin B (AmB), and gramicidin A (GrA) were purchased from Sigma-Aldrich Company Ltd. (Gillingham, UK). KCl solutions (0.1 or 2.0 M) were buffered using 10 mM HEPES-KOH at pH 7.4. Lipids: 1-palmitoyl-2-oleoyl-*sn*-glycero-3-phosphocholine (POPC), 1,2-dipalmitoyl-*sn*-glycero-3-phosphocholine (DPPC), 1-palmitoyl-2-oleoyl-sn-glycero-3-phosphoethanolamine (POPE), 1,2-dioleoyl-*sn*-glycero-3-phosphocholine (DOPC), and cholesterol (CHOL), were obtained from (Avanti Polar Lipids, Inc., USA). All experiments were performed at room temperature (25 °C).

The variations in the chemical structures of the tested chromone-containing allylmorpholines are presented in Table 1. A description of the synthesis of the compounds is given in [7]. Relationships between compound numbers used in this work and in the study by [7] are given in Appendix A. Molecular weights, the ionization constants (*pKa*), logarithms of the octanol/water partition coefficients (*LogP*), and molecular dipole moments (*μ*) of the tested compounds are presented in Appendix A.

### 3.2. Electrophysiological Method for Measuring the Membrane Boundary Potential

Virtually solvent-free planar lipid bilayers were prepared using a monolayer-opposition technique [64] on an aperture with a 50 µm diameter in a 10 µm-thick Teflon film separating the two (*cis*-and *trans*-) compartments of the Teflon chamber. The aperture was pretreated with hexadecane. Lipid bilayers were made from pure POPC. The steady-state conductance of K^+^-nonactin was modulated via the two-sided addition of the chromone-containing allylmorpholines from 10 mg/mL stock solutions in DMSO to the membrane-bathing solution (0.1 M KCl, 5 mM HEPES, pH 7.4) to obtain a final concentration ranging from 10 to 350 μg/mL.

Ag/AgCl electrodes with 1.5% agarose/2 M KCl bridges were used to apply voltage (*V*) and to measure the transmembrane current. “Positive” voltage refers to the case in which the *cis* side compartment is positive with respect to the *trans* side. Current was measured using an Axopatch 200B amplifier (Molecular Devices, LLC, Orleans Drive, Sunnyvale, CA, USA) in the voltage clamp mode. Data were digitized using a Digidata 1440A and were analyzed using pClamp 10.0 (Molecular Devices, LLC, Orleans Drive, Sunnyvale, CA, USA) and Origin 8.0 (OriginLab Corporation, Northampton, MA, USA). Data were acquired at a sampling frequency of 5 kHz using low-pass filtering at 1 kHz, and the current tracks were processed through an 8-pole Bessel 100-kHz filter.

The conductance of the lipid bilayer was determined by measuring the transmembrane current at a transmembrane voltage of 50 mV. The subsequent calculations of the allymorpholine-induced changes in the membrane boundary potential (∆*φ_b_*) were performed according to [39]: (1)$$\frac{G_{m}}{G_{m}^{0}}=\exp(-\frac{q_{e}\Delta \varphi_{b}}{kT}),$$
where *G_m_* and *G_m_*^0^ are the steady-state membrane conductances induced by K^+^-nonactin in the presence and absence of allylmorpholine, respectively; *q_e_* is the electronic charge; *k* is the Boltzmann constant; and *T* is the temperature in Kelvins. It is assumed that the ion concentration in the aqueous phase and the ion mobility within the hydrocarbon region of the bilayer stays the same.

The threshold concentrations of chromone-containing allylmorpholine derivatives in the bathing solutions causing electrical instability and the subsequent destruction of lipid bilayers composed of POPC at *V* = 150 mV in the absence of any other membrane-active compounds were about 500 ± 50 μg/mL. At concentrations lower than the threshold, the tested compounds did not cause an increase in the ion permeability of the membranes (Appendix A).

### 3.3. Dynamic Light Scattering of Liposome Suspensions

Large unilamellar POPC vesicles were prepared via extrusion and were treated with 100 μg/mL of chromone-containing allylmorpholines for 30 min. The control samples were not modified. The hydrodynamic diameters (*d*, nm) and zeta-potentials (ζ, mV) of the lipid vesicles were determined on a Malvern Zetasizer Nano ZS 90 (Malvern Instruments Ltd, Malvern, UK) via the gradual titration of a liposome suspension in PBS at 25 °C.

### 3.4. Fluorimetry of Membrane Dipole Potential

The changes in the membrane dipole potential (∆*φ_d_*) were assessed using a dipole potential-sensitive lipid fluorescence probe, di-8-ANEPPS [65,66]. Large unilamellar POPC vesicles containing 1 mol% di-8-ANEPPS were prepared by extrusion using an Avanti Polar Lipids^®^ mini-extruder (Avanti Polar Lipids, Inc., USA). Chromone-containing allylmorpholines were added to the liposome suspension to a final concentration of 100 μg/mL, and the suspension was incubated for 5 min. The changes in the dipole potential of the POPC bilayers were estimated as described in [67]. Steady-state fluorescence measurements were performed using a Fluorat-02-Panorama spectrofluorimeter (Lumex, Saint-Petersburg, Russia) at room temperature. The fluorescence excitation ratio *R* was defined as the ratio of the fluorescence intensity of di-8-ANEPPS at excitation wavelengths of 420 nm and 520 nm and at the emission wavelength of 670 nm to avoid the influence of the elastic properties of the membrane [67,68]. The measured *R* values were converted into dipole potential values (*φ_d_*) according to the following formula:(2)$$\varphi_{d}(mV)=\frac{R+0.3}{4.3\times 10^{-3}},$$

### 3.5. Electrophysiological Method for Reconstitution of Ion Channels into Planar Lipid Bilayers

Lipid bilayers were made from POPC or a mixture of POPC/CHOL (80/20 mol%) using a monolayer opposition technique [64] and were bathed in 2.0 M KCl at pH 7.4. After the membrane was completely formed and stabilized, a stock solution of GrA and AmB (in ethanol and DMSO, respectively) was added to both sides (GrA) or to the *cis* side (AmB) of the chamber, which were filled up to 1–3 nM and 5–15 μM, respectively. Chromone-containing allylmorpholines were added to both sides of the membranes up to 100 μg/mL.

Conductance of single GrA channels (*g_sc_*) was defined as the ratio of the current flowing through a single GrA pore and *V*. The total number of events used for the channel amplitude analysis was 400–1400. Peaks on the conductance fluctuation histograms were fitted by the normal density function. To verify the distribution hypothesis, the χ2-criterion was applied (*p* ≤ 0.05). To assess alterations in the channel-forming activity of GrA, the product of the number of channels and their probability of being open (*NP_op_*) at *V* = 150 mV was determined using pClamp 10. A ratio of the steady-state transmembrane currents induced by AmB after and before the two-sided addition of the tested agents up to 100 μg/mL (*I_∞_^agent^/I_∞_^control^*) was used to assess the alteration in the pore-forming activity of the polyene macrolide antibiotic at *V* = 50 mV. A brief description of electrophysiological equipment and the parameters of the current records and signal digitization are presented in Paragraph 2.2.

### 3.6. Differential Scanning Microcalorimetry of Liposomal Suspensions

Giant unilamellar DPPC vesicles were prepared via the electroformation method using Vesicle Pre Pro^®^ (Nanion Technologies, Munich, Germany) (standard protocol, 3 V, 10 Hz, 1 h, 55 °C). The resulting DPPC liposome suspension contained 2.5 mM lipids and was buffered by 5 mM Hepes at pH 7.4. A POPE suspension was prepared by dissolving 6–9 mg of dry POPE without or with 0.3–0.45 mg of 7 in a warm buffer (5 mM Hepes, pH 7.4). After that, the suspension was shaken in a 40 °C water bath for 60 min and then sonicated at 40 kHz for 1–3 min. The resulting POPE suspension was kept in the refrigerator 8 °C for at least 8 h before calorimetry measurements. Differential scanning microcalorimetry experiments were performed by a μDSC 7EVO microcalorimeter (Setaram, Caluire-et-Cuire, France). The liposomal suspension was heated and cooled at constant rates of 0.2 and 0.3 °C min^−1^, respectively. The reversibility of the thermal transitions was assessed by reheating the sample immediately after the cooling step from the previous scan. The temperature dependence of the excess heat capacity was analyzed using Calisto Processing (Setaram, Caluire-et-Cuire, France).

The thermograms of DPPC were characterized by the temperature of the pretransition attributed to the mobility of the choline polar head (*T_p_*), the melting temperature (the temperature at which excess heat capacity reaches its maximum, *T_m_*), the enthalpy of the main phase transition (area of the main peak, ∆*H_cal_*), and *T_m_* hysteresis (the difference in the transition temperatures between heating and cooling scans, ∆*T_h_*). The sharpness of the gel-to-liquid-crystalline phase transition was expressed as the temperature difference between the upper (onset) and lower (completion) boundaries of the main phase transition (∆*T_b_*). The changes in the cooperativity of the transition caused by allylmorpholines were also characterized by the ratio of cooperative unit sizes in the absence and presence of the tested compounds (*CUS^control^/CUS^agent^*), determined according to [47]. The main peak decomposition/deconvolution analysis in the presence of chromone-containing allylmorpholines in DPPC liposome suspension was performed using Calisto software. The separation of multiple overlapped peaks was carried out via the application of Gaussian and/or Fraser–Suzuki (asymmetric) signals. The optimal parameters of each peak component, including the percentage contribution to the total area, were determined. The fitting of the calculated signal to the experimental data was performed using non-linear optimization (Marquardt).

The POPE thermograms were characterized by the temperature of the lamellar-to-inverted hexagonal (H_II_) phase transition (*T_HII_*).

### 3.7. Statistical Analysis

The values Δ*φ_b_*, *d*, ζ, Δ*φ_d_*, *g_sc_*, *NP_op_*, *I_∞_^agent^/I_∞_^control^*, Δ*T_p_*, Δ*T_m_*, ΔΔ*H_cal_*, ΔΔ*T_h_*, ΔΔ*T_b_*, *CUS^control^/CUS^agent^,* and *T_HII_* were averaged via 3 to 7 independent experiments and are presented as mean ± standard deviation (*p* ≤ 0.05).

## 4. Conclusions

In summary, we have concluded that:

(i) Chromone-containing allylmorpholines are able to change membrane dipole potential and can be used as dipole potential-modifying compounds in various electrophysiological assays;

(ii) Allylmorpholines modulate the conductance of gramicidin A channels via changing the membrane dipole potential;

(iii) Allylmorpholines affect lipid phase transitions. The effects might be attributed to the allylmorpholine-induced changes in lipid packing, especially to those related to spontaneous curvature stress.

(iv) The ability of allylmorpholines to alter lipid packing is consistent with their potentiating activity on the pore-forming activity of gramicidin A and amphotericin B. Moreover, it is consistent with the antagonistic activity of compounds **4**, **5**, **7**, **10**, and **12** against the NNDA receptors found by [7]. An inhibition of the NMDA-mediated responses by allylmorpholine derivatives might be partially related to the induction of positive curvature stress, similar to lysophospholipids [13].

Thus, chromone-containing allylmorpholines should be considered as general modifiers of the function of different membrane proteins due to their effects on both the electrical and elastic properties of lipid bilayers.

## Figures and Tables

**Figure 1 ijms-23-11554-f001:**
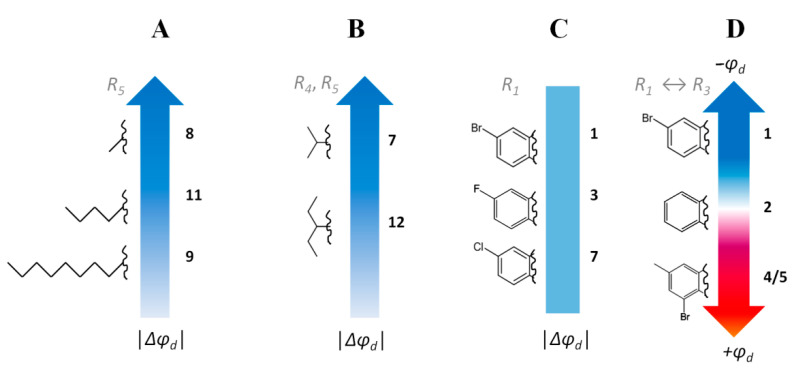
The relationships between the structures of allylmorpholines and their potential-modifying abilities: the dependence of dipole modifying efficiency of allylmorpholines on (**A**) the length of hydrophobic chain in *R_5_* position; (**B**) lengths of hydrocarbon radicals in *R_4_* and *R_5_* positions; (**C**) the type of halogen in the *R_1_* position; (**D**) the existence of halogen in *R_1_* and *R_3_* positions.

**Figure 2 ijms-23-11554-f002:**
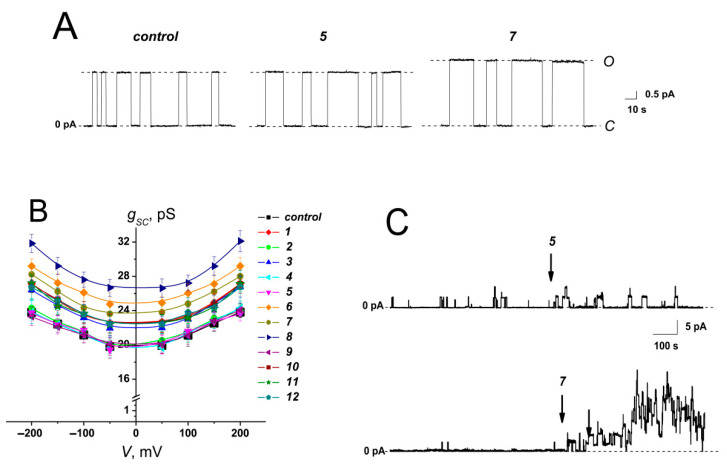
(**A**) Current fluctuations corresponding to opening and closing single gramicidin A channels in the absence (*control*) and presence of **5** and **7** at 100 μg/mL. *V* = 150 mV. *C*—closed state of the channel, *O*—open state of the channel. (**B**) *g_sc_*(*V*) curves of single gramicidin A channels in the absence and presence of chromone-containing allylmorpholines at 100 μg/mL. The relationship between the color of the symbol and the compound is given in the figure legend. (**C**) The effects of **5** and **7** on pore-forming activity of gramicidin A. A peptide was added into the bathing solution at both sides of the bilayers up to 1 nM. The moments when up to 100 μg/mL of **5** (*upper panel*) and **7** (*lower panel*) was added in the bilayer bathing solution are indicated by arrows. The membranes were composed of POPC and bathed in 2.0 M KCl (pH 7.4). *V* = 150 mV.

**Figure 3 ijms-23-11554-f003:**
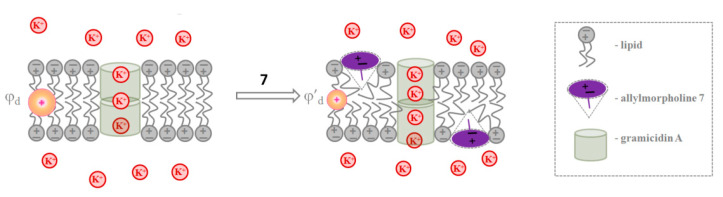
The schematic representations of the mechanisms of the action of allylmorpholine **7** on the conductance of GrA channels via alterations in membrane dipole potential.

**Figure 4 ijms-23-11554-f004:**
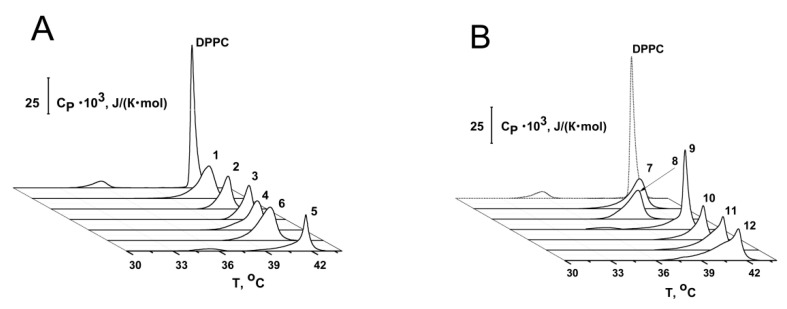
Heating thermograms of DPPC liposomes in the absence and presence of various chromone-containing allylmorpholine derivatives at 100 μg/mL. (**A**) control, **1**, **2**, **3**, **4**, **5**, and **6**; (**B**) **7**, **8**, **9**, **10**, **11**, and **12**. The relationship between the profile and the compound is given in the figure.

**Figure 5 ijms-23-11554-f005:**
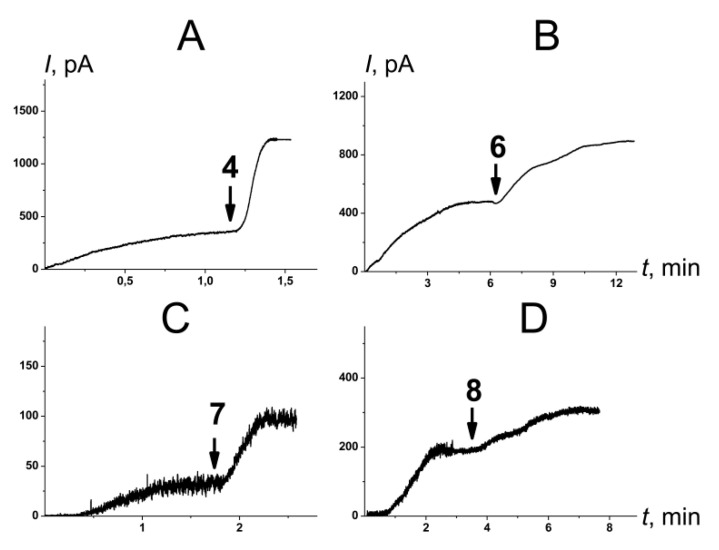
The effects of chromone-containing allylmorpholines (**4**, **6**, **7**, **8**) on the steady-state transmembrane current flowing through membranes modified by one-sided addition of AmB. The moments when 100 μg/mL of **4** (**A**), **6** (**B**), **7** (**C**), and **8** (**D**) was added to the bilayer bathing solution are indicated by arrows. The lipid bilayers were composed of POPC/CHOL (80/20 mol%) and were bathed in 2.0 M 4KCl, pH 7.4. *V* = 50 mV.

**Figure 6 ijms-23-11554-f006:**
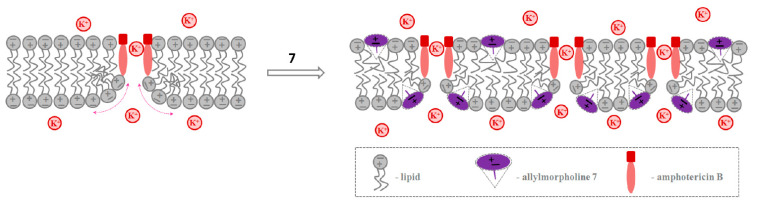
The schematic representations of the mechanisms of action of allylmorpholine **7** on the pore-forming activity of AmB via alterations in lipid-packing stress.

**Table 1 ijms-23-11554-t001:** The characteristic parameters of the action of the tested chromone-containing allylmorpholines on the electrical properties of the POPC membranes and gramicidin A channels at 100 μg/mL.

Agent	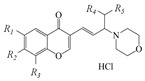					*–*Δ*φ**_b_*, mV	*–ξ*, mV	*–*Δ*φ**_d_*, mV	*g_sc_*, pS	*NP_op_*
	*R_1_*	*R_2_*	*R_3_*	*R_4_*	*R_5_*					
*1*	*-Br*	*-H*	*-H*	*-CH_3_*	*-CH_3_*	105 ± 10	0.1 ± 0.3	106 ± 23	24.9 ± 1.1	2.5 ± 1.3
*2*	*-H*	*-H*	*-H*	*-CH_3_*	*-CH_3_*	26 ± 9	3.4 ± 4.1	–	23.6 ± 1.0	0.3 ± 0.1
*3*	*-F*	*-H*	*-H*	*-CH_3_*	*-CH_3_*	58 ± 10	2.5 ± 0.4	77 ± 25	24.8 ± 0.7	1.1 ± 0.2
*4*	*-CH_3_*	*-H*	*-Br*	*-CH_3_*	*-CH_3_*	−25 ± 7	0.2 ± 0.9	–	22.8 ± 0.9	0.6 ± 0.3
*5*	*-CH_3_*	*-H*	*-Br*	*-(CH_2_)_5−_*	*-(CH_2_)_5−_*	−20 ± 5	0.1 ± 0.3	–	22.7 ± 1.2	0.2 ± 0.1
*6*	*-NO_2_*	*-H*	*-H*	*-CH_3_*	*-CH_3_*	103 ± 11	3.2 ± 0.2	79 ± 25	27.1 ± 1.1	3.1 ± 0.4
*7*	*-Cl*	*-H*	*-H*	*-CH_3_*	*-CH_3_*	110± 14	3.4 ± 0.3	87 ± 23	26.1 ± 1.1	4.1 ± 2.4
*8*	*-Cl*	*-H*	*-H*	*-H*	*-CH_3_*	96 ± 13	0.1 ± 0.2	117 ± 26	29.1 ± 1.1	5.9 ± 3.7
*9*	*-Cl*	*-H*	*-H*	*-H*	*-n* *-C_8_H_17_*	26 ± 12	3.9 ± 0.7	–	22.8 ± 1.1	0.3 ± 0.1
*10*	*-Cl*	*-H*	*-H*	*-(CH_2_)_5−_*	*-(CH_2_)_5−_*	49 ± 9	−0.1 ± 0.1	61 ± 25	24.7 ± 1.7	1.5 ± 0.4
*11*	*-Cl*	*-H*	*-H*	*-H*	*-n*-*C_4_H_9_*	69 ± 13	0.7 ± 1.3	78 ± 24	24.2 ± 1.1	3.9 ± 1.1
*12*	*-Cl*	*-H*	*-H*	*-C_2_H_5_*	*-C_2_H_5_*	54 ± 12	3.5 ± 0.3	60 ± 20	24.3 ± 0.7	1.8 ± 0.2

Δ*φ_b_*, Δ*φ_d_*—the decrease/increase in the POPC–membrane boundary and dipole potential induced by the allylmorpholines. *ξ*—the zeta-potential of POPC vesicles in the presence of allylmorpholines (*ξ*-potential of untreated POPC vesicles was equal to −2.1 ± 0.7 mV). Hydrodynamic diameter of the POPC liposomes in the absence and presence of tested agents was equal to 139 ± 8 and 145 ± 4 nm, respectively, demonstrating that allylmorpholines had no effect on vesicle size. *gsc*—conductance of single GrA channels at *V* = 150 mV in the presence of chromone-containing allylmorpholines in 2 M KCl (pH 7.4) solution bathing POPC bilayers (*gsc* of GrA channels in the absence of allylmorpholines was equal to 22.5 ± 0.9 pS). *NPop*—the product of the number of GrA channels in the POPC membranes and their probability of being open in the presence of the tested compounds at *V* = 150 mV (*NPop* of GrA channels in the absence of allylmorpholines was equal to 0.2 ± 0.1 pS).

**Table 2 ijms-23-11554-t002:** Parameters characterizing the effects of chromone-containing allylmorpholines at 100 μg/mL on thermotropic behavior of DPPC and the pore-forming ability of amphotericin B in POPC/CHOL bilayers.

*Agent*	Δ*T_p_*, °C	Δ*T_m_1_*, °C	ΔΔ*T_b_*, °C	*CUS^control^/CUS^agent^*	*I_∞_^agent^/I_∞_^control^*
*1*	no *	0	2.2 ± 0.2	2.2 ± 0.1	1.1 ± 0.1
*2*	no *	0	2.1 ± 0.5	1.9 ± 0.2	1.1 ± 0.1
*3*	no *	0	1.1 ± 0.1	1.6 ± 0.1	0.8 ± 0.1
*4*	no *	−0.6	3.9 ± 1.4	3.2 ± 1.3	3.4 ± 1.8
*5*	0.8 ± 0.1	0	2.4 ± 0.6	2.4 ± 1.4	1.0 ± 0.1
*6*	no *	−1.0	1.6 ± 0.3	2.0 ± 0.2	1.9 ± 0.3
*7*	no *	−0.5	3.1 ± 0.2	2.5 ± 0.4	2.6 ± 1.6
*8*	no *	−1.9	2.1 ± 0.1	1.6 ± 0.1	1.4 ± 0.4
*9*	0.5 ± 0.1	0	2.4 ± 0.2	1.2 ± 0.2	0.8 ± 0.1
*10*	no *	0	2.1 ± 0.2	1.1 ± 0.1	0.9 ± 0.1
*11*	no *	0	2.7 ± 0.5	1.2 ± 0.1	1.1 ± 0.2
*12*	no *	0	1.8 ± 0.8	1.2 ± 0.2	0.9 ± 0.1

Δ*T_p_*, Δ*T_m_1_*, ΔΔ*T_b_*—the changes in the positions of the pretransition peak and higher melting component and in the width of the main peak in the presence of allylmorpholines. *T_p_*, *T_m_1_*, and Δ*T_b_* of pure DPPC were equal to 34.8, 41.5, and 3.1 °C, respectively. * peak related to pretransition of DPPC was not observed. *CUS^control^/CUS^agent^*—ratio between cooperative unit sizes in the absence and presence of allylmorpholines. *I_∞_^agent^/I_∞_^control^*—ratio between the transmembrane currents induced by AmB in the presence and absence of allylmorpholines. Lipid bilayers were composed of POPC/CHOL (80/20 mol%). Transmembrane voltage was equal to 50 mV.

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
