# Peer review of "Chromone-Containing Allylmorpholines Influence Ion Channels in Lipid Membranes via Dipole Potential and Packing Stress"

_ijms, 2022, doi:10.3390/ijms231911554_

Round 1

Reviewer 1 Report

The authors examine whether a set of 12 allylmorpholine derivatives, which in previous study by co-authors showed pharmacological potential as regulators of neurotransmitter receptors, can partially act through their influence on the lipid microenvironment and regulation of ion channels. For this goal, they incorporated gramicidin A and amphotericin B, both forming ion channels, into model lipid membranes and scrupulously studied the effects of allylmorpholines on electrical and elastic properties of lipid bilayers using a large set of physiochemical methods. Judging by the references to their own works (10 items from all 67 references), the authors are recognized experts in this field of studying lipid bilayers. The manuscript is well written. Nevertheless, there are some points that should be addressed.

1.     Line 59,61. Please change “hydrophilic head” for “hydrophilic headgroup”.

2.     Line 79. “Composes” should be changed for “consists”.

3.     The bathing temperature should be indicated somewhere.

4.     Lines 95,96,97. Please make the narrative more concise.   

5.     Lines 99-103. It is better to remove these three sentences. All data are given in Table 1.

6.     Line 117. Apparently, it would be correct to replace  “the boundary or dipole potentials” with “the surface or dipole potentials”.

7.     Line 130. “We concluded that allylmorpholines did not practically affect zeta-potential of lipid vesicles, and no changes in surface component of the membrane boundary potential had occurred during allylmorpholine adsorption.” Such a conclusion should be made more cautiously and presumably, since the zeta-potential is much weaker than the surface potential.

8.     Line 142. “Measurement error” in statistics is the difference between a measured value of a quantity and its true value. It is better to give a different formulation.

9.     Line 149. We cannot say that we see “a pronounced correlation between …” for all compounds.

10.  Line 246 and Figure 4B. “.. .while 5 and 9 shifted…” There are no compliance with Figure 4B.

11.  Lines 262 and 271. Please explain in the text why a different composition of the lipid bilayer was chosen for the AmB (POPC/CHOL, 80/20 mol%).

12.  Line 310. “..by more than 5°C than introduced…”. It seems, the second “than” should be replaced by “when”.

13. Line 339. Please make a separate section Conclusion.

Author Response

Reviewer 1 general comment.

The authors examine whether a set of 12 allylmorpholine derivatives, which in previous study by co-authors showed pharmacological potential as regulators of neurotransmitter receptors, can partially act through their influence on the lipid microenvironment and regulation of ion channels. For this goal, they incorporated gramicidin A and amphotericin B, both forming ion channels, into model lipid membranes and scrupulously studied the effects of allylmorpholines on electrical and elastic properties of lipid bilayers using a large set of physiochemical methods. Judging by the references to their own works (10 items from all 67 references), the authors are recognized experts in this field of studying lipid bilayers. The manuscript is well written. Nevertheless, there are some points that should be addressed.

Comment 1. Line 59,61. Please change “hydrophilic head” for “hydrophilic headgroup”.

Answer 1. The terminological inaccuracy noted by the Reviewer has been eliminated.

Comment 2. Line 79. “Composes” should be changed for “consists”.

Answer 2. The mistake has been corrected.

Comment 3. The bathing temperature should be indicated somewhere.

Answer 3. All experiments were performed at room temperature (25 °C) (Materials and Methods section, page 10, line 371-372).

Comment 4. Lines 95,96,97. Please make the narrative more concise.

Comment 5. Lines 99-103. It is better to remove these three sentences. All data are given in Table 1.

Answer 4-5. According to the Reviewer's comment, we have revised the paragraph (pages 2-3, lines 97-101).

Comment 6. Line 117. Apparently, it would be correct to replace “the boundary or dipole potentials” with “the surface or dipole potentials”.

Answer 6. We highly appreciate the Reviewer for the careful reading; the unfortunate mistake has been corrected.

Comment 7. Line 130. “We concluded that allylmorpholines did not practically affect zeta-potential of lipid vesicles, and no changes in surface component of the membrane boundary potential had occurred during allylmorpholine adsorption.” Such a conclusion should be made more cautiously and presumably, since the zeta-potential is much weaker than the surface potential.

Answer 7. According to the Reviewer's suggestion, we have rewritten the paragraph (page 4, line 134-138).

Comment 8. Line 142. “Measurement error” in statistics is the difference between a measured value of a quantity and its true value. It is better to give a different formulation.

Answer 8. The inaccuracy noted by the Reviewer has been eliminated.

Comment 9. Line 149. We cannot say that we see “a pronounced correlation between …” for all compounds.

Answer 9. According to the Reviewer's suggestion, we have revised the sentence.

Comment 10. Line 246 and Figure 4B. “.. .while 5 and 9 shifted…” There are no compliance with Figure 4B.

Answer 10. According to the Reviewer's comment, we have rewritten the sentence.

Comment 11. Lines 262 and 271. Please explain in the text why a different composition of the lipid bilayer was chosen for the AmB (POPC/CHOL, 80/20 mol%).

Answer 11. The most widely accepted amphotericin B channel model implies the formation of polyene-sterol complexes that are associated into barrel-type structure (Umegawa et al., Sci Adv., 2022 doi: 10.1126/sciadv.abo2658). Amphotericin B is not able to induce pores in cholesterol-free membranes at μM concentrations (Huang et al., Biophys J. 2002, doi: 10.1016/S0006-3495(02)75326-5). Therefore, we used the cholesterol-enriched membranes to study the effects of allylmorpholines on amphotericin B pore-forming ability. According to the Reviewer's comment, we have revised the paragraph (page 9, lines 326-329).

Comment 12. Line 310. “..by more than 5°C than introduced…”. It seems, the second “than” should be replaced by “when”.

Answer 12. We highly appreciate the Reviewer for the careful reading; the unfortunate mistake has been corrected.

Comment 13. Line 339. Please make a separate section Conclusion.

Answer 13. According to the Reviewer's comment, we have supplemented the manuscript with a separate Conclusion section (page 13, lines 491-510).

Reviewer 2 Report

In the manuscript, the authors reported the effects of chromone-containing allylmorpholines on ion channels model lipid membranes. Several allylmorpholines were employed to observe such effects, where the boundary dipole potential and lipid packing stress were taken as observation parameters. The authors concluded that chromone-containing allylmorpholines should be considered as general modifiers of the function of different membrane proteins due to their effects on both the electrical and elastic properties of lipid bilayers. The manuscript is well organized and the results are meaningful. Thus, I recommend it to be published in International Journal of Molecular Sciences after minor revision. My concern is only the following minor issues:

1.     The authors are better to discuss the formation of pores in membranes in more detail.  

2.     Why the authors choose such a model membrane?

Author Response

Reviewer 2 general comment.

In the manuscript, the authors reported the effects of chromone-containing allylmorpholines on ion channels model lipid membranes. Several allylmorpholines were employed to observe such effects, where the boundary dipole potential and lipid packing stress were taken as observation parameters. The authors concluded that chromone-containing allylmorpholines should be considered as general modifiers of the function of different membrane proteins due to their effects on both the electrical and elastic properties of lipid bilayers. The manuscript is well organized and the results are meaningful. Thus, I recommend it to be published in International Journal of Molecular Sciences after minor revision. My concern is only the following minor issues.

Comment 1. The authors are better to discuss the formation of pores in membranes in more detail.

Answer 1. According to the Reviewer’s comment, the Results and Discussion section has been substantially supplemented (page 5, lines 185-187, 188-192, 208-209; page 9, lines 326-329, 349-351).

Comment 2. Why the authors choose such a model membrane?

Answer 2. Phosphocholine is abundant lipid of mammalian cell membranes (Harayama and Riezman, Nat. Rev., 2018 doi.org/10.1038/nrm.2017.138). It is widely used to mimic the membranes of eukaryotic cells, and there is a lot of information about its physicochemical properties (Parchekani et al., Sci. Rep., 2022 doi.org/10.1038/s41598-022-06380-8). POPC is neutral at physiological conditions.

The estimation of boundary, surface and dipole potentials of membranes, and electrophysiological registration of gramicidin A channels was made using POPC. g(V)-curves of single gramicidin A channels and the number of gramicidin A channels in the presence of all tested allylmorpholines were also studied in the POPC bilayers.

POPC cannot be used for DSC measurements due to its low melting point about -2°C (https://avantilipids.com/tech-support/physical-properties/phase-transition-temps). For this reason, PC with two saturated palmitoyl tails (DPPC) with phase transition temperature about 41°C was used. Cone-shaped POPE was chosen to study the effect of derivative 7 on the transition from lamellar to inverted hexagonal phase (ΔTHII of POPE is about 70°C) to confirm an assumption about the influence of the agent on the membrane curvature stress.

The most widely accepted amphotericin B channel model implies the formation of polyene-sterol complexes that are associated into barrel-type structure (Umegawa et al., Sci Adv., 2022 doi: 10.1126/sciadv.abo2658). Amphotericin B is not able to induce pores in cholesterol-free membranes at μM concentrations (Huang et al., Biophys J. 2002, doi: 10.1016/S0006-3495(02)75326-5). Therefore, we used POPC/CHOL (80/20 mol%) membranes for studying the effects of allylmorpholines on the pore-forming activity of amphotericin B. Nevertheless, despite some differences in the lipid systems used for calorimetric measurements of the disordering action of allylmorpholines on membrane lipids and their effects on amphotericin B channels, Figure 5, Supplementary Figure 5 and Table 2 clearly demonstrate that only compounds 4, 6, 7, and 8 having the greatest disordering effect on DPPC and shifting its main transition peak to lower temperatures, caused an increase in the pore-forming activity of amphotericin B in POPC/CHOL bilayers (increasing polyene-produced steady-state transmembrane current).